# A reliance on human habitats is key to the success of an introduced predatory reptile

Tom Major[1,2]*, Lauren Jeffrey[1], Guillem Limia Russel[1], Rebecca Bracegirdle[1], Antonio Gandini[1], Rhys Morgan[1], Benjamin Michael Marshall[3], John F. Mulley[1], Wolfgang Wüster[1]

1 School of Environmental and Natural Sciences, Bangor University, Bangor, Gwynedd, United Kingdom,
2 Department of Life & Environmental Sciences, Bournemouth University, Poole, Dorset, United Kingdom,
3 Department of Biological and Environmental Sciences, University of Stirling, Stirling, Scotland, United Kingdom

* tmajor@bournemouth.ac.uk

**Data Availability Statement:** All data and code are available from https://figshare.com/s/10b8bafb3ce0069555a1

## Abstract

Understanding the success of animals in novel environments is increasingly important as human-mediated introductions continue to move species far beyond their natural ranges. Alongside these introductions, inhabited and agricultural areas are spreading, and correspondingly most animal introductions occur in populated areas. Commensal species which can live alongside humans by making use of specific conditions, structures, or prey, have a significant advantage. Introduced mammal species often use anthropogenic features in their environment and demonstrate a higher tolerance of human disturbance, but their importance remains understudied in ectotherms. The Aesculapian snake (*Zamenis longissimus*) is an ectotherm which has been introduced beyond the northern extremities of its natural range. To understand their persistence, we radio-tracked snakes daily over two active seasons, including high-frequency tracking of a subset of males. We investigated snake home range size using Autocorrelated Kernel Density Estimators (AKDE). Using AKDE-weighted Habitat Selection Functions we identified preferences for habitat features in a mosaic of habitats, and we used Integrated Step Selection Functions to further explore how these features influence movement. We revealed a particular preference for buildings in male snakes, while females preferred woodland. We demonstrate that the success of this ectothermic predator is likely tied to a willingness to use human features of the landscape.

## Introduction

Through time, the areas inhabited by species are pushed and pulled in many directions by climate, habitat changes, interactions with other species, and human transport. The latter has become a particularly powerful force, introducing species to distant, novel habitats where they are exposed to a range of entirely different physical and climatic processes as well as different biotic interactions to those in their native ranges. Not all species are equally successful when faced with such pressures. Identifying the key characteristics underpinning the success or failure of species in novel environments can help predict winners and losers in a future where all

**Funding:** This work was supported by the Knowledge Economy Skills Scholarships (KESS II – case number 80815) and the Welsh Mountain Zoo, awarded to WW and JM for TM's PhD. Additional funding was provided by the Biodiversity and Ecosystem Evidence and Research Needs (BEERN) Programme awarded to TM.

**Competing interests:** The authors have declared that no competing interests exist.

animals face increased challenges [1]. Through the tracking of animal movements, we can understand traits which are key to success in unfamiliar locations, such as colonisation and dispersal abilities, and habitat requirements [2–4]. Using these insights, we can begin to explore how introduced species adapt to and impact their new habitats.

Many introduced species demonstrate a willingness to utilise human features, including fragmented landscapes and anthropogenic structures, to travel and hunt [5], and have generally demonstrated greater tolerance of human disturbance than their native counterparts [6, 7]. Using anthropogenic features is advantageous because non-native species are most likely to be introduced in or near human-impacted habitats, and it is frequently generalist species that capitalise on these disturbed habitats and make good invaders [8]. While much of the available literature focuses on terrestrial endotherms and their adaptability in the face of human dominated landscapes, comparatively little focuses on ectotherms–particularly snakes [9]. This is despite several prominent examples of introduced snakes (albeit in more natural landscapes) having considerable impacts on the native fauna [10, 11].

Snake introductions are not limited to warm climates. Aesculapian snakes (*Zamenis longissimus* (Laurenti, 1768)) are a constricting species in the family Colubridae. Native to mainland Europe, they range from France in the West to Iran in the East [12]. The species previously occupied a larger part of Northern Europe, only recently going extinct in Denmark during the early 1900s [13], with remaining relict populations in Germany, Switzerland, the Czech Republic, and Poland. Aesculapian snakes have been introduced twice to the UK in modern times [13]. Snakes were introduced to Colwyn Bay, North Wales when an unknown number of individuals escaped from the Welsh Mountain Zoo in the 1970s due to structural failure of their enclosure. This population represents the northernmost modern population of Aesculapian snakes, and its persistence here raises the question of how a species can expand northward, despite the constraints of ectothermy.

Over several decades, and especially since the advent of surgically implanted transmitters [14], radiotelemetry has become the backbone of snake spatial ecology. However, despite technological and methodological advancements, few studies have capitalised on recent advances in techniques and analyses to explore the full potential of telemetry data to illuminate the spatial ecology of reptiles. Much of the literature still uses outdated techniques and analyses to calculate habitat selection and home range size [15]. Home range estimation measures such as minimum convex polygons and kernel density estimators have been continued to be used in determining the home range of animals [15]. These methods assume independent and identically distributed data, and do not account for autocorrelation, where data points close in time are also close in space [16]. This was generally acceptable for studies of animals which exhibit frequent movement, especially because of the large time lag between points that result from manually relocating the animal using VHF telemetry. However, for studies of animals which spend long periods inactive, such as snakes, autocorrelation presents a major concern and renders traditional home range estimators unreliable [15]. Therefore we incorporated the novel approach of Autocorrelated Kernel Density Estimation (AKDE), which overcomes the limitations of traditional estimation techniques in that it accounts for autocorrelation, and these models present the additional benefit of providing confidence intervals for the estimates of home range size provided [17]. Another continued limitation in most telemetry studies of reptiles involves short tracking durations and infrequent tracking schedules, often ranging from once-daily to weekly tracking. Here, we included a two-hour daily tracking schedule in order to not only distinguish home range sizes more accurately but also to discern dispersal habitat selection. Using an advanced analysis called Integrated Step Selection Functions (ISSF) enabled the modelling of movement and habitat choice simultaneously.

The overall aim of this study was to investigate the spatial ecology of an introduced preda-tor, adapted to a warmer climate but existing in cool North Wales. We used radiotelemetry to determine how these animals use their new range, and we set out to discover the home range and space use requirements of both male and female Aesculapian snakes. Our second goal was to discover the habitat preferences of this species, which is crucial to understanding its survival. As we had experienced snakes entering buildings and vegetation piles, we hypothesised that snakes may be reliant on human features of the landscape. Finally, we were keen to know the dispersal capabilities of this species. We wanted to learn which habitats represent pathways to mobile snakes, allowing us to infer their likely routes should this population spread into the surrounding area. We hypothesised that hedgerows, as linear features in the habitat, would represent pathways for snakes travelling longer distances.

## Methods

### Study area and animals

The study site (approximately 1.72 km$^2$, see S1 Fig) is located between the town of Colwyn Bay and the village of Mochdre, North Wales, UK (53.28–53.29˚N, -3.74–3.76˚W). The area con-sists of a mosaic of habitats, including housing, with meadows and pastures separated by hedgerows. Most pastures are grazed periodically by sheep and cattle. The site also includes the Welsh Mountain Zoo covering an area of 0.15 km$^2$. The Zoo grounds are highly disturbed because of the constant maintenance involving the removal of trees or vegetation, landscaping or building of structures. There is also a patch of woodland in the north corner. Most of the animal faecal matter and vegetation waste is transported to a large dung heap in the southeast-ern corner of the zoo. Patches of deciduous forest and small patches of gorse scrub are scat-tered over the entire study site, and a small patch of woodland to the east of the Zoo was in the process of being removed for a new development in 2022. Roads are found throughout, with busy roads surrounding the zoo and the connecting meadows and pastures, with a dual car-riageway (A55) at the site's northernmost extremity.

We implanted 21 adult Aesculapian snakes with radio transmitters during June–October 2021 and May–September 2022 (see S1 and S2 Tables for tracking and capture summary). Our sample consisted of 13 males and eight females. Snakes were caught by hand, either during dedicated surveys, opportunistically, during radio-tracking activities, or following notification by members of the public. Two tracked individuals were caught by keepers at the Welsh Mountain Zoo during their daily activities. Because of the difficulty of finding Aesculapian snakes, which in this population takes approximately eight hours of searching per adult snake found (unpublished data), we radio-tracked any available adult snakes with sufficient body diameter to carry a transmitter, which was approximately 20 mm at the beginning of the poste-rior third of the snake excluding the tail. Snakes were transported to Bangor University for transmitter application, and we collected morphometric data including snout-vent length (SVL), tail length (TL) and mass (S2 Table). We attempted to minimise the time snakes were kept in captivity for implantation procedures (n = 21 implantations, mean = 7.34 ± 6.7 days held, range = 1–23 days). Snakes were held in 70L plastic boxes (710 x 545 x 190 mm) in a tem-perature-controlled room at 21˚C with suitable refuge and water provided *ad libitum*.

Depending on their size, snakes were implanted with 1.2, 1.4 or 1.6 g Holohil BD-2T radio transmitters (Holohil Inc, Canada) following Reinert and Cundall [14], using isoflurane anaes-thetic and butorphanol analgesia, with an internal securing stitch [18]. Post-implantation, snakes were kept overnight at a constant 21˚C and released the following day. One snake (F159) was held for an additional day post-surgery to ensure wound closure. Snakes were released at their exact point of capture in dry conditions warmer than 14˚C. The only

exception was M180 who was caught basking on top of a hedge in a residential garden. The homeowner requested we release the snake a short distance from their garden, and we released him approximately 30 m away in a hedgerow. As snakes appeared to be behaving normally immediately after release, we began tracking the following day and did not discard any data.

## Radiotelemetry

We employed two different tracking regimes. Because Aesculapian snakes are primarily diurnal [19], we tracked them only during daylight hours. To assess longer distance movements and home range sizes, snakes were located twice daily as part of two daily tracking rounds, the first beginning at 10:00 and a second at 14:00. As we were a small team tracking many snakes, the timings of the tracks were not precise, but animals were usually tracked once in the morning and once in the afternoon, and almost always twice daily. See S2 and S3 Figs for tracking periods and time lags.

In 2022 a subset of seven male snakes were located five times daily to determine what type of habitat the snakes use when they are moving, with the greater tracking frequency allowing us to capture more points in the movement pathway. These snakes were tracked in sessions beginning at 09:00, 11:00, 13:00, 15:00 and 17:00. One female (F203) was tracked five times daily between 14/05/2022 and 14/06/2022, before switching to twice daily. After 20/08/2022, we began tracking the four male snakes with remaining transmitter battery twice daily. When we located a snake, we recorded the temperature, humidity, and location, and noted any behaviour. We used a Garmin GPSMAP 64S to collect GPS locations. To avoid disturbing the snake and thus influencing their behaviour, we attempted to keep 10 metres between observers and tracked snakes, but within the confines of narrow gardens this was not always possible.

## Snake home range estimation

To implement Autocorrelated Kernel Density Estimations (AKDE), we used the *ctmm* package [20] and R v4.2.1 [21] to fit continuous time stochastic process movement models to our snake movement data. We first checked individuals for range residency to ensure their range is no longer expanding by calculating the semi-variance function and visualising it using variogram analysis [22]. We removed individuals that did not demonstrate range-residency from the home range analysis. We fitted multiple movement models and used AICc to identify the model best fitting the autocorrelation structure for each snake. These were either the Ornstein-Uhlenbeck (OU) model where the animal exhibits Brownian motion restricted to a finite home range, or the OUF model with continuous-velocity motion restricted to a finite home range, or Independent Identically Distributed (IID). These prototype models are either isotropic, where there is equal diffusion in every direction, or anisotropic, where diffusion varies depending on direction [20]. For optimal performance we estimated autocorrelation and covariance bias using perturbative hybrid residual maximum likelihood (pHREML), accounting for both small absolute sample size and small effective sample size [23]. Absolute sample size refers to the total number of times the animal was located during the tracking period, while effective sample size is the entire tracking duration divided by how long it takes, on average, for the animal to cross its linear home range. We then fit AKDEs using the guidance provided by [17]. We used weighted AKDEc and pHREML to estimate home range size, which reduces oversmoothing of range limits, particularly in cases with small effective sample sizes [17]. It also helps to address irregular sampling. Silva et al. [17] recommend pHREML and AKDEc for effective sample sizes (range crossings) below 20, which applied to most individuals. Parametric bootstrapping can also be used to reduce estimation error, and we used ctmm.

boot in the *ctmm* package to apply parametric bootstrapping to individuals with low effective sample sizes [24, 25].

## Space use

In recent years there has been a rise in the use of dynamic Brownian Bridge Movement Models (dBBMMs) to estimate the home ranges of animals, but these estimators are not suitable for this purpose [26]. As these models are occurrence distribution estimators and not range distribution estimators, these models use animal locations over time to interpolate where animals might have been during a tracking period, rather than extrapolating to their entire range [26]. However, dBBMMs have utility in estimating space use during the study period, and allow for comparison with other studies, giving a valuable impression of snake spatial ecology during the study period. As many snakes were unlikely to have sufficient tracking durations to facilitate home range estimation, we use estimates of space use derived from dBBMMs.

We estimated snake space use using the R package *move* v4.1.10 [27]. As snakes were generally active for a few days before spending between a few days and a week inactive, we specified a moving window of 11. With our twice daily tracking regime, this allowed us to detect variations in the behavioural state of snakes over a six day period [28]. We chose a margin size of three to allow detection of active vs inactive states [28], and used the mean GPS error of our snake locations as the telemetry error. We used two daily tracks to model space use across all individuals.

## Habitat selection

All analyses were conducted using R v4.2.1 [21]. For all habitat use analyses, we used custom shape files of habitat types created in QGIS Using modified code from Smith et al. [29] and Hodges et al. [30], we converted our raster layers into layers with continuous gradients denoting the Euclidean distance to habitat features. These layers were then inverted to avoid zero-inflation and to ensure that the resulting outputs were easy to interpret, as positive values indicate selection.

## Integrated resource (habitat) selection functions

Traditional resource selection function analyses, now termed habitat selection functions [31], do not allow for autocorrelated data and assume independence between each point at which an animal is located. However, for high-resolution GPS location frequencies, or for animals which move infrequently, points are not independent from each other. To avoid the need to thin data to ensure independence of points, we utilised habitat selection functions informed by our AKDEs, which down-weight autocorrelated points rather than discarding them [32]. We used rsf.fit within the *ctmm* package [22] to fit integrated resource selection functions to our snake tracking data, with simultaneously estimated spatial constraints. We used the Monte Carlo numerical integrator for likelihood evaluation, with a numerical error threshold of 0.05 for the parameter estimates and log-likelihood.

## Integrated step selection functions

We used Integrated Step Selection Functions (ISSF) to analyse how habitat types influenced the movements of Aesculapian snakes at the population scale. This allowed us to incorporate the movements of all tracked individuals, even those with short tracking durations that did not demonstrate range-residency. We split the data into male and female snakes and adapted code from Smith et al. [29] and Muff et al. [33] to run mixed conditional Poisson regression models

essentially operating as "population-level ISSFs" on our twice-daily tracking data. The first and third daily tracks of snakes which had been tracked five times daily were used to ensure comparability with our twice daily data. Using the *INLA* package v22.05.07 [34] we ran a population-level ISSF to ascertain either association or avoidance of our eight habitat classifications. We removed zero distance steps (non-moves) from this analysis. We generated 200 random points per move, for step length we used a Gamma distribution and for turn angle Von Mises [29, 30], to facilitate comparison between locations the snakes selected and locations they did not. We created eight single-factor models, one for each habitat type, and all models included the interaction of turn angle and step length. As per Muff et al. [33] the stratum-specific random effect of step was set to 0.0001. In keeping with Smith et al. [29], we utilised a Penalised Complexity prior, PC (1, 0.05) for the random slope, which was individual, and uninformative normal priors were used for the fixed effects. Using the *INLA* package v22.05.07 [34], we used nested Laplace approximations in fitting these models.

For the male snakes tracked five times daily, we ran both individual ISSFs as well as the "population-level ISSFs" described above to investigate whether Aesculapian snakes demonstrated attraction to or avoidance of habitat types at the scale of the whole population. For individual selection, we created ISSFs using *INLA* package v22.05.07 [34], filtering steps using a resample rate of two hours with a tolerance of four hours to avoid bias from unexpected overnight movements. We created nine single-factor models, one for each habitat type, and one representing the null model. The individual habitat models also included the interaction of turn angle and step length, while the null model only included the interaction of step length and turn angle. Otherwise, we used the same settings as described above. We used AIC scores to determine which features most strongly influenced the habitat selection of Aesculapian snake individual movements from our ISSF. Models with the lowest AIC score or scores < 2 greater than the lowest were considered to have the largest influence.

## Seasonality and shelter locations

We used two measures to determine seasonal behaviour changes in Aesculapian snakes, mean daily displacement and dBBMM-derived motion variance [30]. We use dBBMMs to visualise the space use of tracked snakes over the tracking period and ascertain areas of high use and reuse that are of particular significance to Aesculapian snakes. We also use the estimates of motion variance they provide to determine periods of heightened activity during the tracking period [30].

Although mean daily displacement (MDD) has limitations, we were unable to estimate speed or distance travelled using continuous-time movement models [10], likely due to the relative infrequency of our data points compared to high-resolution GPS data. Despite the shortcomings of MDD [35], we kept to a strict tracking schedule of at least two tracks per day for all individuals, allowing increased confidence in the MDD estimates. To investigate seasonality, we filtered the first and third daily tracks of snakes which received five tracks per day to allow comparison between all individuals. We summed the movement distance across these two daily tracks to create our values for daily displacement. We opportunistically collected observational data on the breeding behaviour of this population during tracking activities, which allowed us to better inform the dates of the mating and egg-laying seasons.

To understand areas snakes used for long periods, we first created move objects in R using twice daily tracking data for all 21 individuals with the package *move* v4.1.10 [27]. Using the *recurse* package v1.1.2 [36], we set a radius of five meters around each location a snake visited. We recorded the amount of time each snake spent in the same location for multiple fixes as time spent stationary. Finally, we used the *recurse* package to visualise places the snakes had

spent long periods, and recorded what these locations were by overlaying the GPS locations on a map. We verified the locations using our behavioural notes.

### Limitations and data statement

In line with the STRANGE framework [37], we recognise that there may be bias in the trapp-ability and self-selection of our sample of snakes. Snakes were frequently near people or their dwellings when captured. Seven of the 13 male snakes were captured at different times under one tarpaulin covering a wood pile in the garden of a residential home, while three of eight females came from one garden containing multiple mature compost heaps. As many snakes were found near each other, there is also a higher likelihood they are related.

All figures were created using R v4.2.1 (https://r-project.org/). For data manipulation we used the *amt* package v0.1.7 [38] and the *dplyr* package v1.0.10 [39]. For data visualisation we used *ggplot2* v3.3.6 [40]. For visualising movement data and creating tracks we used *move* v4.1.10 [27]. All data and code used in this study are available from the online repository Fig-share (https://doi.org/10.6084/m9.figshare.25817770.v1). This work took place with approval from the Bangor University Animal Welfare and Ethics Review Board. Telemetry procedures were conducted under Home Office project licence PC44793B6.

## Results

### Movements

We tracked 13 adult male and eight adult female Aesculapian snakes between June and October 2021 and May and September 2022. The average tracking duration was 56.67 ± SE 7.85 days (range 4–126 days). We collected a total of 3232 snake locations, which included 947 relo-cations (a move from the previous location). The mean daily displacement (MDD) for six male snakes on the twice daily tracking regime was 38.45 ± 21.33 m, and for the seven males on the five times daily regime it was 52.34 ± 26.43 m. The MDD of eight females tracked twice daily was 26.14 ± 18.55 m.

### Snake home ranges

Eight males and three females were found to have stable home ranges using variogram analysis (see S4 and S5 Figs). The remaining ten individuals did not reach an asymptote [41]. For these individuals, the short tracking durations meant that there was insufficient information to inform our home range estimation model, and they were excluded from Autocorrelated Ker-nel Density Estimation (AKDE). Full model results for all individuals can be found in Table 1. The mean effective sample size for snakes included in the AKDE analysis was 12.92 ± 8.57 (range 4.29–35.57). These figures are relatively low and demonstrate that using the pHREML method and weighting the AKDEs was justified [17]. The mean 95% AKDE estimate for home range size for the three range resident females was 23.32 ± 29.74 ha (range 0.28–65.31 ha). For eight range resident males it was 28.86 ± 28.42 ha (range 2.05–92.16 ha). For range resident snakes (Fig 1), the top models were Ornstein-Uhlenbeck (OU) or Ornstein-Uhlenbeck forag-ing (OUF). Two individuals, F159 and M218, had low effective sample sizes of six and 4.3 respectively, with pHREML bias in the order of 3% or higher (Table 1), and we applied parametric bootstrapping to these individuals. All models reflected elliptical home ranges (anisotropic) except for F158 which was more circular (isotropic). The traditional KDE approach of using Independent Identically Distributed (IID) points proved ineffective and the IID models had high dAICc values (S3 Table).

**Table 1. Results of the snake AKDE home range analysis.** Effective sample size refers to the number of times the animal crossed its home range during the tracking period, while absolute sample size is the total number of observations (fixes). Contour area estimates of home range for individuals presented in hectares. F159 and M218 had parametric bootstrapping applied to reduce error on their home-range estimates and the estimates displayed are post-bootstrapping.

| ID | AICc top model | Effective sample size | Absolute sample size | 95% AKDE lower CI (ha) | 95% AKDE estimate (ha) | 95% AKDE upper CI (ha) | pHREML bias | Parametric bootstrap bias |
|---|---|---|---|---|---|---|---|---|
| F158 | OU | 8.72 | 102 | 1.97 | 4.37 | 7.72 | 0.013 | - |
| F159 | OUF anisotropic | 4.71 | 102 | 20.10 | 64.75 | 135.03 | 0.028 | 0.0096 |
| F177 | OU anisotropic | 35.57 | 126 | 0.19 | 0.28 | 0.38 | 0.001 | - |
| M031 | OU anisotropic | 19.44 | 165 | 1.24 | 2.05 | 3.05 | 0.003 | - |
| M137 | OU anisotropic | 9.36 | 209 | 9.98 | 21.44 | 37.20 | 0.011 | - |
| M139 | OU anisotropic | 8.69 | 167 | 7.83 | 17.41 | 30.75 | 0.013 | - |
| M154 | OU anisotropic | 7.66 | 302 | 4.54 | 10.75 | 19.59 | 0.017 | - |
| M180 | OU anisotropic | 14.89 | 96 | 1.46 | 2.61 | 4.10 | 0.005 | - |
| M202 | OU anisotropic | 8.83 | 457 | 23.37 | 51.57 | 90.74 | 0.013 | - |
| M209 | OU anisotropic | 18.74 | 423 | 19.72 | 32.89 | 49.37 | 0.003 | - |
| M218 | OU anisotropic | 4.08 | 192 | 100 | 120.04 | 261.50 | 0.054 | 0.0147 |

## Snake space use

We estimated space use using dBBMMs (Table 2). We excluded three snakes with tracking durations less than 14 days (F212, F219, and M073) from the following means as their space use estimates would likely skew the results. For females, the mean area of the 95% confidence occurrence distributions was 2.09 ± 3.79 ha (range = 0.02–10.51 ha). The mean area of the 95% confidence occurrence distributions for males was 6.34 ± 7.10 ha (range 0.46–20.72 ha).

## Individual habitat selection via AKDE-weighted RSF

We had sufficient data to inform range-residency and perform the habitat selection analysis on three female and eight male Aesculapian snakes. These parameter estimates for resource selection are visualised in Fig 2. The results for the remaining snakes can be found in S6 Fig and S4 Table. In male snakes, buildings were the most commonly selected-for habitat type with five of eight individuals demonstrating positive selection for buildings and two more strongly suggesting it without definitive evidence (Fig 2). M137 and M139 showed a preference for woodland, while M202 and M209 were associated with pasture. One chose gardens (M180), one selected meadows (M202), and similarly only (M154) showed a preference for hedgerows. For females, F159 showed avoidance of pasture. We were unable to determine individual preference or for any habitat type for females in this analysis. With regards to males avoiding habitat types, three individuals (M031, M180 and M202) avoided roads, while M202 avoided woodland.

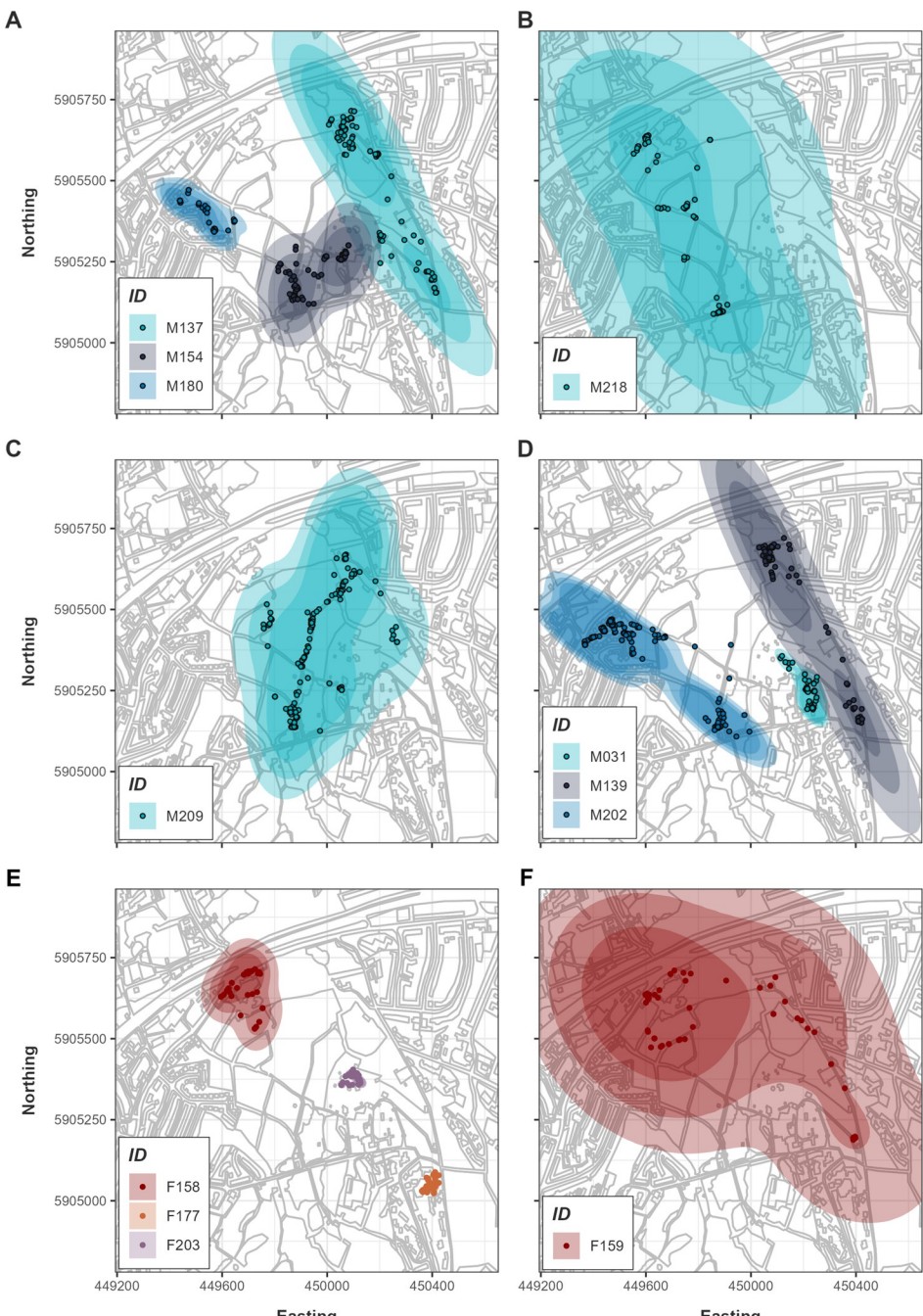

**Fig 1. AKDE home ranges for tracked snakes that demonstrated range-residency.** A-D) Male snakes. E, F) Female snakes. Darkest shading in the centre represents the lower confidence interval for the 95% home range contour, medium shading is the 95% contour, and the lightest shading is the upper confidence interval. Points represent location data of each animal. While F203 did not demonstrate range residency, we plot the data here for illustration purposes as she was tracked for a relatively long period of 65 days.

## ISSF at the population scale

The ISSF analysis suggests woodland is of importance for females at the population-level (Fig 3). Males demonstrated habitat generalism, showing selection for meadows, scrub, road

**Table 2. dBBMM occurrence distributions and mean daily displacement (MDD) for the Aesculapian snakes tracked in this study.**

| ID | 90% (ha) | 95% (ha) | 99% (ha) | Tracking duration (days) | Total distance moved (m) | MDD (m) | Maximum daily distance (m) |
|---|---|---|---|---|---|---|---|
| F050 | 0.02 | 0.05 | 0.16 | 28 | 216.20 | 7.74 | 56.78 |
| F142 | 0.02 | 0.02 | 0.04 | 33 | 264.26 | 8.05 | 58.09 |
| F158 | 0.55 | 1.30 | 3.23 | 50 | 1028.29 | 20.57 | 92.10 |
| F159 | 5.91 | 10.51 | 18.42 | 61 | 3673.04 | 60.38 | 364.43 |
| F177 | 0.21 | 0.32 | 0.59 | 90 | 782.51 | 8.68 | 44.67 |
| F203 | 0.19 | 0.32 | 0.92 | 65 | 1282.96 | 19.70 | 72.06 |
| F212 | 0.07 | 0.09 | 0.13 | 4 | 157.93 | 38.04 | 64.73 |
| F219 | 0.01 | 0.01 | 0.02 | 12 | 555.29 | 46.00 | 244.47 |
| M031 | 0.63 | 1.03 | 1.97 | 33 | 1135.56 | 34.20 | 129.11 |
| M073 | 9.03 | 12.48 | 20.82 | 9 | 698.28 | 78.71 | 206.39 |
| M074 | 0.01 | 0.01 | 0.02 | 30 | 179.99 | 5.97 | 64.96 |
| M137 | 1.82 | 4.51 | 14.49 | 126 | 3894.85 | 30.95 | 382.88 |
| M139 | 2.27 | 7.21 | 22.72 | 95 | 3903.90 | 41.10 | 559.30 |
| M149 | 1.56 | 2.76 | 6.41 | 37 | 1706.27 | 46.37 | 336.90 |
| M154 | 1.79 | 3.67 | 7.57 | 74 | 2427.55 | 32.70 | 107.16 |
| M178 | 0.29 | 0.46 | 0.87 | 83 | 999.82 | 12.04 | 255.31 |
| M180 | 1.31 | 2.13 | 4.48 | 57 | 1225.27 | 21.54 | 125.12 |
| M202 | 5.55 | 10.75 | 25.43 | 112 | 7935.06 | 70.85 | 566.25 |
| M209 | 14.10 | 21.00 | 35.59 | 107 | 7387.04 | 69.03 | 365.16 |
| M217 | 0.35 | 1.85 | 5.27 | 17 | 1477.92 | 86.94 | 383.95 |
| M218 | 11.94 | 20.72 | 43.71 | 67 | 4469.91 | 66.73 | 586.54 |

surfaces, and possibly hedgerows, but appearing to select areas nearer to buildings and gardens most strongly (Fig 3). We were unable to detect differences in step length associated with different habitat types for either females or males at the population scale (S7 and S8 Figs).

## Step length from five times daily tracking

We included the data from six of the seven males that were tracked five times daily, excluding M074 who spent 29 of 30 tracking days stationary. We investigated the influence of habitat type on the step lengths of individual snakes (Fig 4). For M202, shorter steps were associated with buildings and roads. For M218, both proximity to pasture and hedgerows were associated with shorter steps.

We also determined which features influenced the habitat selection of snakes. The top model was pasture for M218. Gardens were top or second for M154 and M217. Hedgerows were top or second for M154 and M209, while M207. M031, M154, and M202 were influenced by buildings (S9 and S10 Figs and S5 Table). Model selection results do not necessarily equate to definite selection or avoidance, however. At the individual level, two snakes, M154 and M202, preferred to be closer to meadows and pasture. M218 showed a positive association with hedgerows and scrub. None of the remaining males tracked five times daily showed definitive habitat association in this analysis. We also ran the population-level analysis for this group, who demonstrated a preference for hedgerows, buildings and scrub, but we found no significant influence of habitat type on step length using the population-level analysis with this group (S11 Fig).

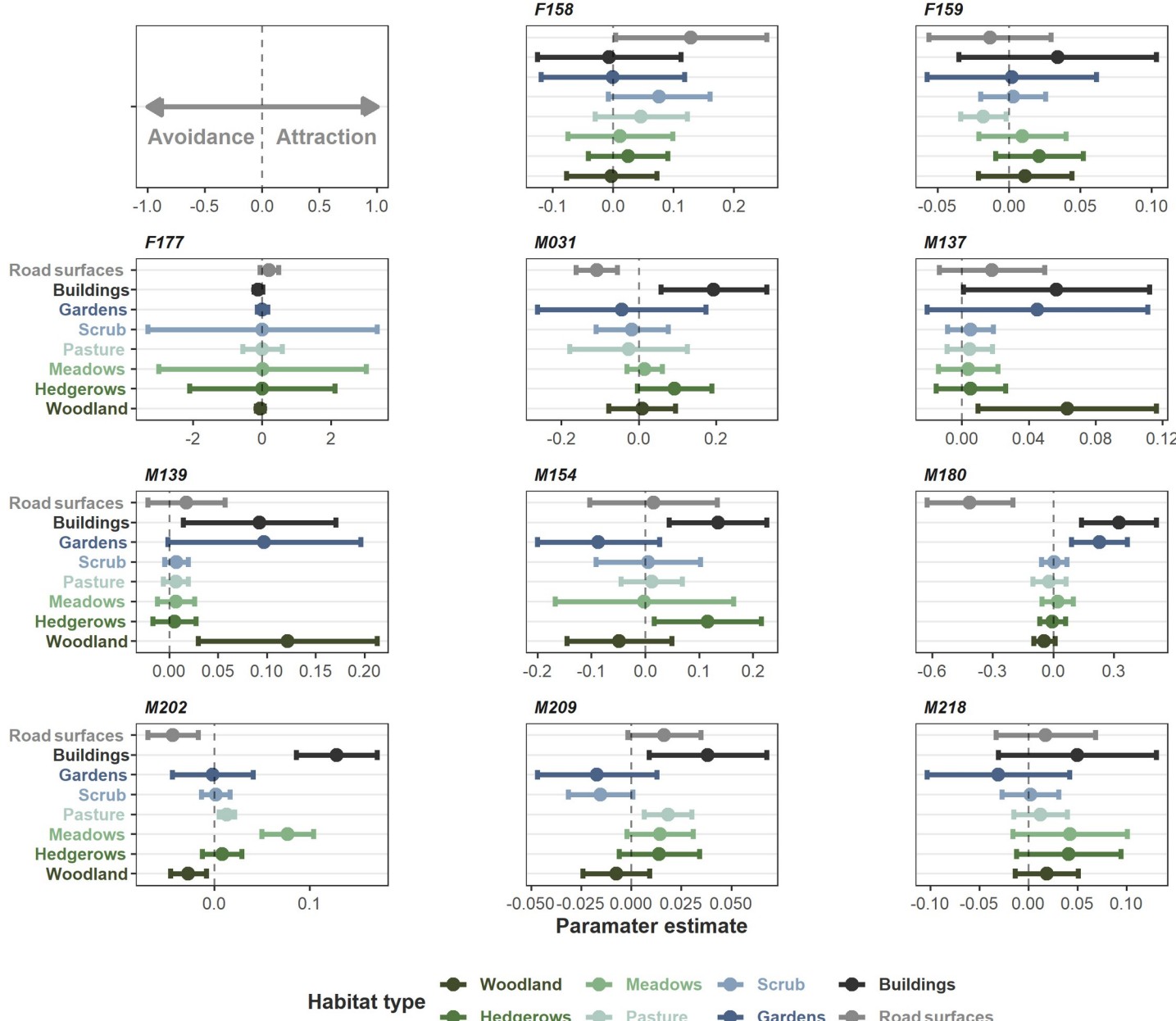

**Fig 2. Coefficients from the weighted AKDE habitat selection functions for range resident snakes.** Each plot displays the habitat selection of one individual snake. Positive values indicate selection for a habitat type, while negative values indicate avoidance. Error bars represent 95% confidence intervals.

## Seasonality

Female movement showed a visible peak in motion variance during the egg-laying season from July to mid-August (Fig 5). The mean daily displacement (MDD) of females during the egg-laying season was 34.05 ± 75.45 m/day compared to 12.52 ± 32.33 m/day during the rest of the year (Fig 6). Across all females there was only one day where an individual moved > 100 m outside of the egg-laying season, and females were sedentary outside of this period. We discovered eggs inside the compost heap of a residential property immediately after F177 left it on 10/08/2021. F159 made an uncharacteristically long 364m move in mid-July that we interpret

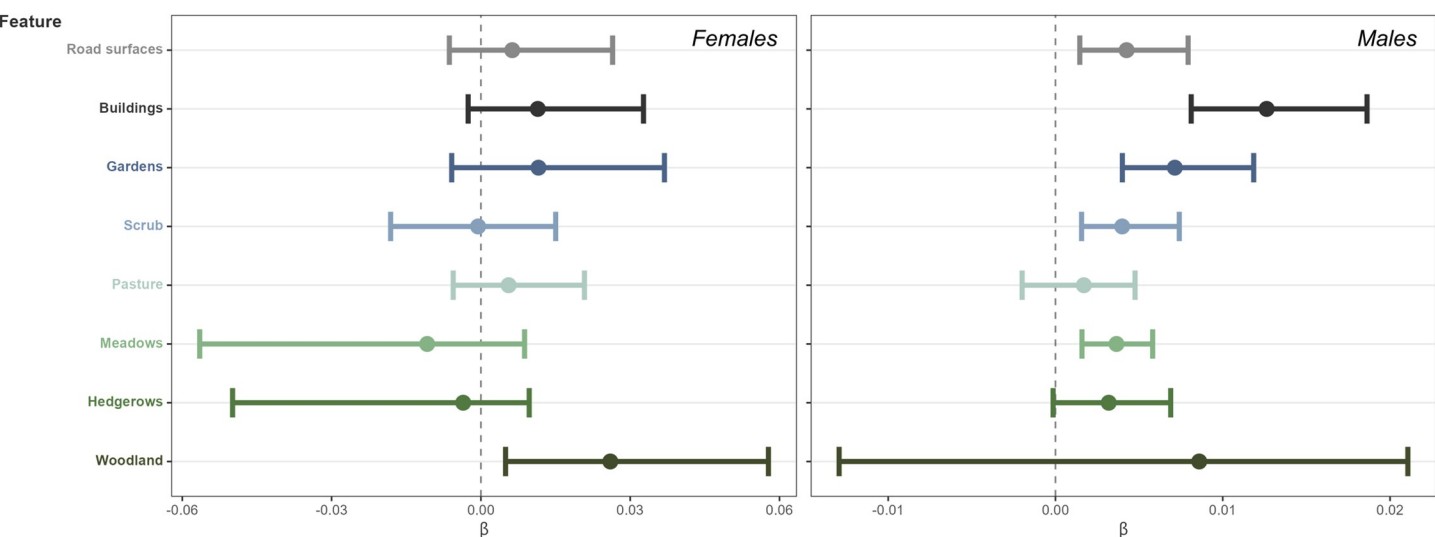

**Fig 3. Results of the ISSF analysis at the population-level for female snakes (n = 8) and male snakes (n = 13).** Bars represent 95% confidence intervals. Positive values indicate selection toward a particular habitat type, while negative values indicate avoidance.

**Fig 4. Individual integrated step selection functions demonstrating the interaction between step length and habitat types for Aesculapian snakes under the five times daily tracking regime.** Positive values indicate longer steps associated with a habitat feature, and negative values indicate shorter steps. Error bars indicate 95% confidence intervals. Circles indicate the features which were included in models with the lowest AIC score or scores less than two higher.

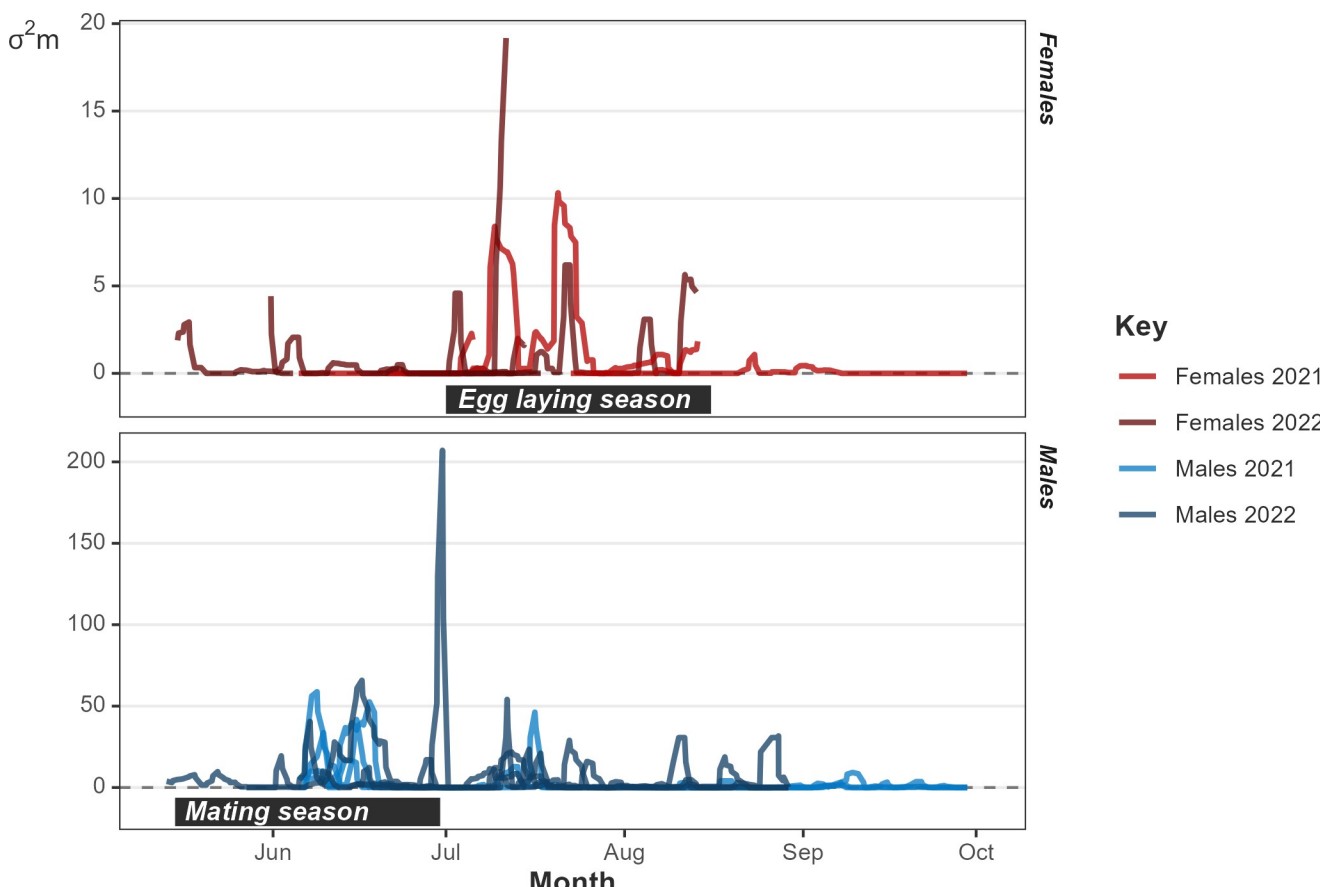

**Fig 5. Motion variance plot for females and males through the study periods in 2021 and 2022.** Peaks represent increased movement distances.

as nest searching, before spending three days (13/07/2021–16/07/2021) inside the dung pile within the Welsh Mountain Zoo. While we could not locate her eggs despite extensive search, this may be due to the size of the pile, which is approximately 10 m across. We found a separate clutch of eggs inside this pile in 2019, confirming that it is an egg-laying site for the species. One further Aesculapian snake egg was found by Zoo staff in a pile of wood chippings 20 m from the dung heap on the Zoo site in 2021.

The males showed a visible peak in motion variance the mating season between mid-May and the end of June (Fig 5). The MDD of males during the mating season was 68.7 ± 112.22 m/day compared to 26.3 ± 58.39 m/day during the rest of the year (Fig 6). Males only moved distances > 500 m per day during the mating season, despite occasional periods of high activity in mid-summer. We observed one tracked male mating on 13/06/2021 and another on 24/05/2022. We witnessed one of our tracked males in combat with an unknown male on 15/06/2022. These observations reinforce our understanding of the timings of the mating season of this species in Wales.

## Snake mortality

Of our sample of 21 tracked snakes, five died during their tracking periods, and one died after tracking was completed. In total, three females died. F177 was tracked during 2021 until the transmitter ran out of battery, and was then found dead the following year on 15/08/2022 after a car strike (see S2 Table). F142 was also killed by a car strike on 07/07/2021 after being tracked

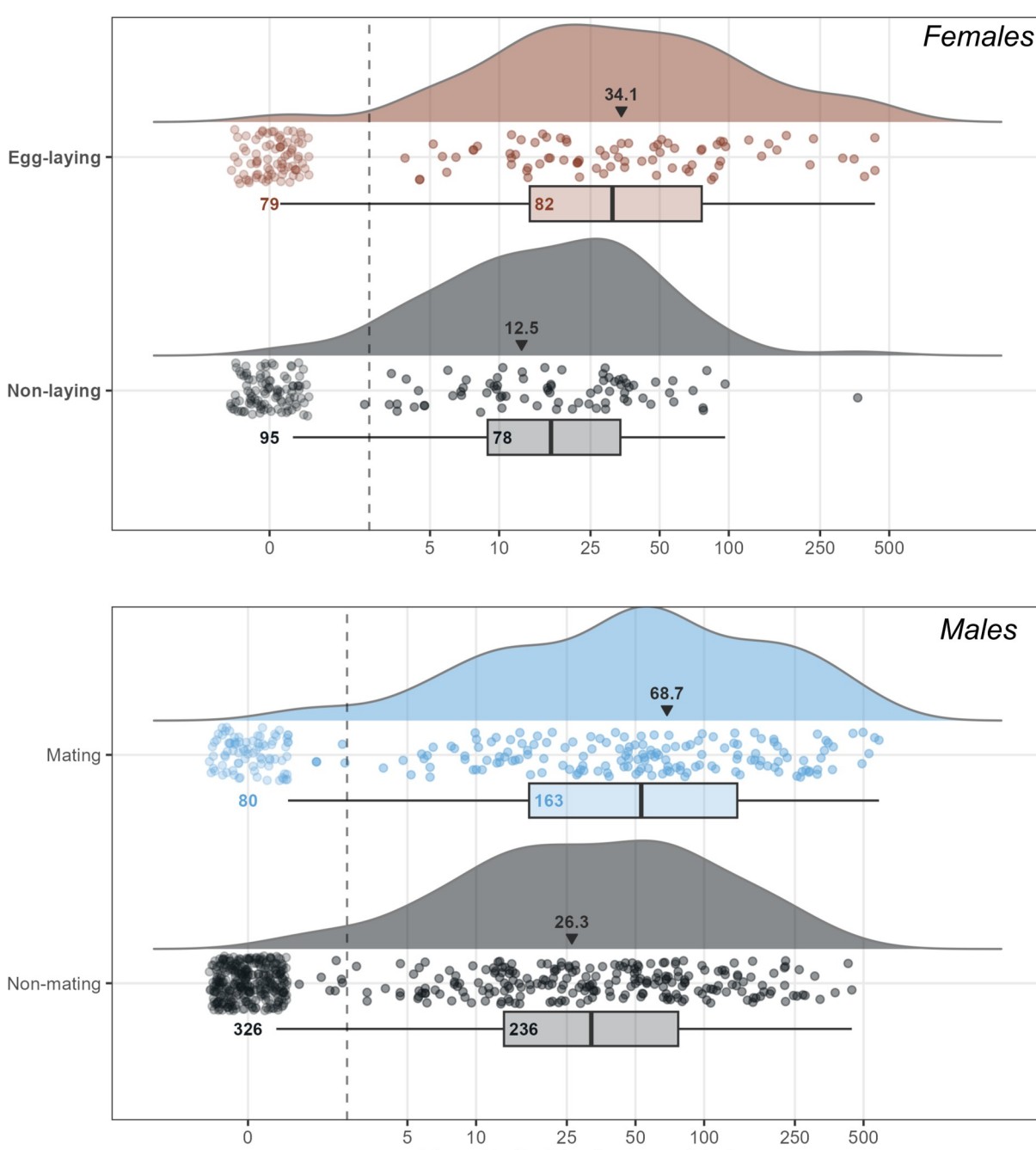

**Fig 6. Raincloud plot visualising the mean daily displacement of eight tracked female Aesculapian snakes and 13 tracked male Aesculapian snakes through 2021–2022.** Each point represents the distance moved by a snake on an individual day. Non-moves are excluded from the box and density plots, with numbers inside the box showing the number of moving days in each season. Days without movement are plotted to the far left along with a count. Mean daily displacement for each season, with non-moves included, are printed along with a small black arrow. The egg-laying season (July 1st–August 16th) and the mating season (15th May– 30th June) are compared with the rest of the active season for females and males respectively. Figure created using code adapted from [30].

for 33 days, on the same two-laned stretch of road as F177. Both snakes were gravid at the time of death, containing four and seven well-developed eggs respectively. Lastly, F159 was cannibalised by a tracked male (M137) in August 2021 [42].

Three of 13 male snakes died during the study. In 2021, M073 was predated by buzzards, with his transmitter signal discovered in a pine tree containing a buzzard nest, approximately 30 m high up. M149 died from mammal predation. The snake was found at the edge of a meadow with approximately one metre of trampled grass in all directions with a broken neck. Small sections of the snake had been consumed. We suspect either a stoat, badger, or domestic cat was responsible. Finally, in the 2022 season, M031 was killed by a car strike on the entrance road within the Welsh Mountain Zoo grounds.

## Time spent sheltering

All snakes spent time stationary, being located at the same location for at least two tracks in succession. The mean time spent stationary of all females was 4.91 ± 4.04 days (0.5–14.27 days). For all males, the mean time spent stationary was 5.14 ± 6.67 days (range 0.82–27.61 days). Generally, when snakes were in the same location for multiple successive fixes, they were inside shelter. The exception were two snakes who were repeatedly found in basking sites in vegetation at the edges of roads (M137 and M218). Females selected a road verge (n = 1), compost or vegetation piles (n = 4), and buildings (n = 3) as repeated or long-term shelter. Males chose a road verge (n = 1), compost or vegetation pile (n = 1), or buildings (n = 10). Five males used two different buildings for shelter, while three males used three different buildings. These shelter sites were often used for long periods, and seven males and seven females spent ≥ 10 days in an individual shelter (S6 Table).

## Discussion

Our suite of habitat and step selection analyses reveal that male Aesculapian snakes show a distinct preference for buildings in their introduced range in North Wales, with seven of eight individuals in our AKDE-weighted habitat selection function selecting buildings as habitat, and our population SSF model highlighted their importance to male snakes. Like many introduced species, Aesculapian snakes were introduced close to an urban area following escape from captivity. Urban areas often contain under-utilised resources, and introduced species which can capitalise on them have a significant advantage [43]. For snakes, anthropogenic structures such as buildings and culverts provide shelter, thermoregulatory opportunities and egg-laying sites [30, 44–47], and Aesculapian snakes are known to use man-made structures in the northern part of their native range [19]. Snakes in our study frequently took shelter in human features of the habitat, and compost heaps, vegetation piles, and buildings represented long-term shelter for female snakes. Females spent long periods inactive during the spring and early summer, remaining in shelter - particularly compost heaps. Males frequently used buildings as shelter sites, with one individual using a hole in a road verge and one a vegetation pile as long-term or revisited shelter sites. As snakes in this population prey predominantly on small mammals, supplemented with birds (unpublished data), these locations may provide foraging opportunities in addition to shelter and thermoregulatory benefits.

We observed Aesculapian snakes actively seeking and returning to use inhabited buildings, and climbing large structures to access the attics and wall cavities of houses. This attraction towards anthropogenic features is unusual behaviour compared with native snake species in the UK that often avoid built-up urban areas. The Adder (*Vipera berus*) and smooth snake (*Coronella austriaca*) are rarely found in human-dominated environments. Grass snakes (*Natrix* sp. including *Natrix helvetica*) can be found in anthropogenic environments and will use compost heaps, manmade structures, or ponds in gardens, but their use of anthropogenic features is significantly less extensive [48–50]. With urbanisation continuing unabated, the ability to survive in human-dominated landscapes is valuable to individual species and vital for

maintaining biodiversity [51]. Herpetofauna and invertebrates remain understudied in this realm, and our suite of analytical approaches represents an unusually detailed investigation into the lives of a cryptic reptile species.

Our evidence here suggests that, like other snakes, Aesculapian snakes have peaks in activity due to their reproductive cycles [52, 53]. The activity of male snakes peaked in the mating season in May and June, coinciding with observations of male-male combat and mating, while females exhibited a definite peak during summer when they travel to lay their eggs. We discovered eggs in rotting vegetation built up by humans - a compost heap, a pile of wood chippings, and a dung heap at the Welsh Mountain Zoo, further demonstrating the importance of human elements of the habitat to this introduced species.

Despite their ability to benefit from human features of the landscape, the semi-rural range of this introduced predator also contains dangers. Three of 21 tracked snakes died to road mortality. Road mortalities can be high for juvenile Aesculapian snakes [54], and our study site lacks any culverts beneath the roads, which are known to be utilised by snakes for road crossing [55]. Indeed, road mortality was found to be low for adult Aesculapian snakes in a population where culverts allowed safe passage under roads [54]. Two females died on roads while heavily gravid, potentially suggesting an increased risk of road mortality when travelling to lay eggs. Reptiles are known to increase their movement distances in highly disturbed areas, despite keeping smaller home ranges [56], further exacerbating the risk of road mortality. Three snakes demonstrated avoidance of roads in this study, suggesting they may be responsible for limiting the expansion of this species, although this is more likely the case for wider roads [57, 58]. Two other mortalities in this study were caused by a mammalian predator and a buzzard, and one was cannibalised by an Aesculapian snake [42].

## Conclusion

In summary, this study demonstrates the importance of human-dominated habitats to an introduced predator. While Aesculapian snakes are present in the fossil record of the UK, they had been absent for likely 300,000 years [59]. Their impact on native fauna remains uncertain, but they coexist with similar assemblages elsewhere, particularly in western mainland Europe. Worldwide, animal ranges are shifting poleward or to higher elevation as the climate warms dramatically because of human activity [60]. The UK is now home to an increasing number of mobile species which can travel over sea from further South, including numerous moths and butterflies [61], and wetland birds [62]. Aesculapian snakes have similarly migrated northward, only via human transport instead of natural means. That said, it seems likely that North Wales represents the northernmost tolerance limit of this species in current climatic conditions. The use of buildings for shelter and vegetation piles for egg laying appear to be important to their success in a temperate climate that is further north than any remaining native populations. However, simply being successful in an area is not evidence to suggest the area contains ideal conditions [63], and the broad range of habitats selected by individual snakes in this study suggest that Aesculapian snakes are adaptable generalists, capable of using mixed habitat and unafraid of using buildings and other features in close proximity to humans.

## Supporting information

**S1 Fig. Map of the study area in Colwyn Bay showing the habitat classifications used in the habitat selection analyses.** Inset map shows location of the study site in Wales, UK, with Conwy County highlighted red. Map created using QGIS [64].
(TIF)

**S2 Fig.** Tracking periods for 21 tracked snakes over (A) 2021 and (B) 2022. Each point represents a tracking occasion where the snake was located, with a higher density of points representing increased tracking frequency.
(TIF)

**S3 Fig.** Time lag between tracks for snakes which were tracked A) twice daily and B) five times daily. Both show a first peak between the tracks during a day, and a second which represents the gap overnight. The checked line marks the mean time lag between tracks.
(TIF)

**S4 Fig. Variogram plots visualising the semi-variance function for tracked female snakes.** F158, F159 and F177 can be said to have reached range residency as the plots reach an asymptote and stop rising.
(TIF)

**S5 Fig. Variogram plots visualising the semi-variance function for tracked male snakes.** M031, M137, M139, M154, M180, M202, M209 and M218 can be said to have reached range residency as the plots reach an asymptote and stop rising.
(TIF)

**S6 Fig. Coefficients from the weighted AKDE habitat selection functions for snakes which were not range resident.** Each plot displays the habitat selection of one individual snake. Positive values indicate selection for a habitat type, while negative values indicate avoidance. Error bars are 95% confidence intervals. As these snakes were not range resident, we cannot have confidence in values for their parameter estimates.
(TIF)

**S7 Fig. Interaction between step length and habitat types for females (n = 8) in the ISSF analysis at the population-level.** Error bars represent 99% confidence intervals.
(TIF)

**S8 Fig. Interaction between step length and habitat types for males (n = 13) in the ISSF analysis.** These results are from two daily tracks of all individuals.
(TIF)

**S9 Fig. Individual integrated step selection functions for distance to habitat features for Aesculapian snakes (*Zamenis longissimus*) under the five times daily tracking regime.** Positive values indicate a positive association with a habitat feature. Error bars indicate 95% confidence intervals. Circles indicate the features which were included in models with the lowest AIC score or scores < 2 greater.
(TIF)

**S10 Fig. Distance to habitat features at the population-level for male Aesculapian snakes on the five times daily tracking regime, excluding M074 (n = 6).** Positive values indicate a positive association with a habitat feature. Error bars indicate 95% confidence intervals.
(TIF)

**S11 Fig. Interaction between step length and habitat features at the population scale for six male Aesculapian snakes on the five times daily tracking regime.** Error bars indicate 95% confidence intervals.
(TIF)

**S12 Fig.**
(JPG)

**S13 Fig.**
(JPG)

**S14 Fig.**
(JPG)

**S1 Table. Capture records and time in captivity for radio-tracked Aesculapian snakes in this study, with additional movement metrics, and relocations.** MDD = mean daily displacement.
(DOCX)

**S2 Table. Tracking and capture summary for all 21 tracked snakes.** Abbreviated capture types are mating with a tracked snake (mating), notification from the public (notification), and dedicated survey (survey). We did not record snout-vent length (SVL) for F203 in error.
(DOCX)

**S3 Table. Model fitting and selection results from the AKDE home range analysis.** Model abbreviations are as follows: OU = Ornstein-Uhlenbeck, OUF = Ornstein-Uhlenbeck foraging process, IID = independently and identically distributed. dRMSPE is the root mean squared prediction error. DOF area is the effective sample size.
(DOCX)

**S4 Table. Results of the AKDE-weighted resource (habitat) selection function for all individuals.** Positive values denote selection for a particular habitat type, while negative values denote avoidance.
(DOCX)

**S5 Table. Model selection results from the ISSF analysis of males which were tracked five times daily in 2022.** Asterisks mark the top performing model based on AIC scores and models with an AIC score < 2 from the top model. All models included the association of habitat with step length and turn angle.
(DOCX)

**S6 Table. Time spent stationary for all 21 radio-tracked Aesculapian snakes.**
(DOCX)

## Acknowledgments

We are indebted to the staff of the Welsh Mountain Zoo for their tireless support and assistance. Thank you also to the generous residents of Colwyn Bay who allowed us to conduct fieldwork on their properties with enthusiasm. We thank Henry Crisp, Jack Pearce, Gem Shinasi, and Andrea Pozzi for volunteering their time to radio-track snakes. Thank you to Becca Snell for assisting with radio transmitter implantations. Finally, we thank Dr Damian Lettoof and one anonymous reviewer for their reviews of the manuscript.

## Author Contributions

**Conceptualization:** Tom Major, Benjamin Michael Marshall, John F. Mulley, Wolfgang Wüster.

**Data curation:** Tom Major, Lauren Jeffrey.

**Formal analysis:** Tom Major, Lauren Jeffrey, Benjamin Michael Marshall.

**Funding acquisition:** Tom Major, John F. Mulley, Wolfgang Wüster.

**Investigation:** Tom Major, Lauren Jeffrey, Guillem Limia Russel, Rebecca Bracegirdle, Antonio Gandini, Rhys Morgan, Benjamin Michael Marshall.

**Methodology:** Tom Major, Benjamin Michael Marshall, John F. Mulley, Wolfgang Wüster.

**Project administration:** John F. Mulley, Wolfgang Wüster.

**Resources:** Wolfgang Wüster.

**Software:** Tom Major.

**Supervision:** John F. Mulley, Wolfgang Wüster.

**Validation:** Tom Major.

**Visualization:** Tom Major, Lauren Jeffrey, Benjamin Michael Marshall.

**Writing – original draft:** Tom Major, Lauren Jeffrey, Guillem Limia Russel, Rebecca Bracegirdle, Antonio Gandini, Rhys Morgan, Benjamin Michael Marshall, John F. Mulley, Wolfgang Wüster.

**Writing – review & editing:** Tom Major, Lauren Jeffrey, Guillem Limia Russel, Rebecca Bracegirdle, Antonio Gandini, Rhys Morgan, Benjamin Michael Marshall, John F. Mulley, Wolfgang Wüster.

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
