## [Decision Letter · Decision Letter 0]

15 Nov 2024

PONE-D-24-37502A reliance on human habitats is key to the success of an introduced predatory reptilePLOS ONE

Dear Dr. Major,

Thank you for submitting your manuscript to PLOS ONE. After careful consideration, we feel that it has merit but does not fully meet PLOS ONE’s publication criteria as it currently stands. Therefore, we invite you to submit a revised version of the manuscript that addresses the points raised during the review process.

We look forward to receiving your revised manuscript.

Kind regards,

Sonny Shlomo Bleicher, Ph.D.

Academic Editor

PLOS ONE

Journal Requirements: When submitting your revision, we need you to address these additional requirements. 1. Please ensure that your manuscript meets PLOS ONE's style requirements, including those for file naming. The PLOS ONE style templates can be found at https://journals.plos.org/plosone/s/file?id=wjVg/PLOSOne_formatting_sample_main_body.pdf and https://journals.plos.org/plosone/s/file?id=ba62/PLOSOne_formatting_sample_title_authors_affiliations.pdf 2. Thank you for stating the following financial disclosure: "This work was supported by the Knowledge Economy Skills Scholarships (KESS II – case number 80815) and the Welsh Mountain Zoo, awarded to WW and JM for TM's PhD. Additional funding was provided by the Biodiversity and Ecosystem Evidence and Research Needs (BEERN) Programme awarded to TM."  Please state what role the funders took in the study.  If the funders had no role, please state: ""The funders had no role in study design, data collection and analysis, decision to publish, or preparation of the manuscript."" If this statement is not correct you must amend it as needed. Please include this amended Role of Funder statement in your cover letter; we will change the online submission form on your behalf. 3. Please expand the acronym “KESS II and BEERN” (as indicated in your financial disclosure) so that it states the name of your funders in full. This information should be included in your cover letter; we will change the online submission form on your behalf. 4. Please note that in order to use the direct billing option the corresponding author must be affiliated with the chosen institute. Please either amend your manuscript to change the affiliation or corresponding author, or email us at plosone@plos.org with a request to remove this option. 5. When completing the data availability statement of the submission form, you indicated that you will make your data available on acceptance. We strongly recommend all authors decide on a data sharing plan before acceptance, as the process can be lengthy and hold up publication timelines. Please note that, though access restrictions are acceptable now, your entire data will need to be made freely accessible if your manuscript is accepted for publication. This policy applies to all data except where public deposition would breach compliance with the protocol approved by your research ethics board. If you are unable to adhere to our open data policy, please kindly revise your statement to explain your reasoning and we will seek the editor's input on an exemption. Please be assured that, once you have provided your new statement, the assessment of your exemption will not hold up the peer review process. 6. Please include your full ethics statement in the ‘Methods’ section of your manuscript file. In your statement, please include the full name of the IRB or ethics committee who approved or waived your study, as well as whether or not you obtained informed written or verbal consent. If consent was waived for your study, please include this information in your statement as well. 7. Please include captions for your Supporting Information files at the end of your manuscript, and update any in-text citations to match accordingly. Please see our Supporting Information guidelines for more information: http://journals.plos.org/plosone/s/supporting-information. 8. Please review your reference list to ensure that it is complete and correct. If you have cited papers that have been retracted, please include the rationale for doing so in the manuscript text, or remove these references and replace them with relevant current references. Any changes to the reference list should be mentioned in the rebuttal letter that accompanies your revised manuscript. If you need to cite a retracted article, indicate the article’s retracted status in the References list and also include a citation and full reference for the retraction notice.

Additional Editor Comments:

Dear Tom Major et al.

The reviewers found the manuscript to be sound and only requires relatively small edits.

Please make those corrections as suggested by the reviewers and address the queries they made.

Sorry for the delay, but securing reviewers can be challenging these days.

Thanks for your patience.

Best wishes,

Sonny Bleicher

Reviewers' comments:

Reviewer's Responses to Questions

**Comments to the Author**

1. Is the manuscript technically sound, and do the data support the conclusions?

Reviewer #1: Yes

Reviewer #2: Yes

2. Has the statistical analysis been performed appropriately and rigorously? 

Reviewer #1: Yes

Reviewer #2: Yes

3. Have the authors made all data underlying the findings in their manuscript fully available?

Reviewer #1: Yes

Reviewer #2: Yes

4. Is the manuscript presented in an intelligible fashion and written in standard English?

Reviewer #1: Yes

Reviewer #2: Yes

5. Review Comments to the Author

Reviewer #1: The research paper is well-written and technically sound. It provides detail and insightful statistical analysis on studying a cryptic ectotherm. Against the backdrop of climate change and global trade, including the exotic reptile trade, research such as presented in this paper is essential to understanding the ecological drivers behind establishment success, and greatly adds to our knowledge.

I have been made aware of several reptile and amphibian escape events in the United Kingdom and the Republic of Ireland, and this paper provides detailed methodology on studing alien species. It will provide a firm base on which future research in cool climates can be undertaken. It is hoped that in future radio-telemetry studies can be undertaken on smaller ectotherms as technology develops and perhaps within fully-automated real-time systems. This paper provides additional data on succesful radio-telemetry studies.

Reviewer #2: This is great succinct study showcasing movement and space-use of a relatively large, introduced snake to Wales, UK.

The authors have conducted impressive effort to hand radio-tracking a difficult to assess taxon and have good sample sizes considering the study system. They have employed the best tools for assessing movement and space use.

I only have minor comments:

Lines 125-126: Is there a reference for this or perhaps just specify where this data comes from?

Lines 127: Body diameter over what size approximately?

Discussion: I noticed there is no mention of Aesculapian snake prey species or what they are preying on in this introduced range? As this is a major component of driving species persistence, habitat use, and potential impacts to local fauna it is probably worth a mention somewhere in the discussion?

Further, given the general global focus of introduced predators/species is environmental impact it seems odd not to mention a perspective on this.

6. PLOS authors have the option to publish the peer review history of their article (what does this mean?). If published, this will include your full peer review and any attached files.

Reviewer #1: No

Reviewer #2: **Yes: **Damian C Lettoof

---

## [Author Response · Author response to Decision Letter 0]

2 Dec 2024

Response to reviewers: PONE-D-24-37502, Major et al

Thank you very much for the thorough review and consideration of our work. We have added the following elements based on editors’ comments:

General comments 

• Changed corresponding author affiliation to Bangor University (1) and Bournemouth University (2) which reflects the fact that TM conducted this work during his PhD in Bangor.

• Added amended role of funder statement to cover letter.

• Added link to Figshare data and code repository, and ethics statement, in ‘Limitations and data statement’ of methods.

• Made the data and code available and no longer embargoed.

• Corrected the supporting information section of the manuscript and the separate supporting information file.

Comments to reviewers:

Reviewer 1

• Clarified how the snakes escaped from the Welsh mountain Zoo on line 70, “Snakes were introduced to Colwyn Bay, North Wales when an unknown number of individuals escaped from the Welsh Mountain Zoo in the 1970s due to structural failure of their enclosure.”

• Line 147, clarified Aesculapian snakes are primarily diurnal.

• Line 450, added line to discussion mentioning use of man-made structures in the northern part of the range and cited Kovar et al 2016.

• Made all minor grammatical changes suggested in the marked up PDF, both in text and references.

Reviewer 2

Lines 125-126: Is there a reference for this or perhaps just specify where this data comes from?

• Added explanation that the time taken to find snakes comes from our own unpublished data

Lines 127: Body diameter over what size approximately?

• Added explanation, “we radio-tracked any available adult snakes with sufficient body diameter to carry a transmitter, which was approximately 20 mm at the beginning of the posterior third of the snake excluding the tail.”

Discussion: I noticed there is no mention of Aesculapian snake prey species or what they are preying on in this introduced range? As this is a major component of driving species persistence, habitat use, and potential impacts to local fauna it is probably worth a mention somewhere in the discussion?

• Added a line at 457: “As snakes in this population prey predominantly on small mammals, supplemented with birds (unpublished data), these locations may provide foraging opportunities in addition to shelter”.

Further, given the general global focus of introduced predators/species is environmental impact it seems odd not to mention a perspective on this.

• Added a line in the conclusion, “While Aesculapian snakes are present in the fossil record of the UK, they have been absent for likely 300,000 years (Ashton et al., 1994). Their impact on native fauna remains uncertain, but they coexist with similar assemblages elsewhere, particularly in western mainland Europe.”

---

## [Decision Letter · Decision Letter 1]

20 Dec 2024

A reliance on human habitats is key to the success of an introduced predatory reptile

PONE-D-24-37502R1

Dear Dr. Major,

We’re pleased to inform you that your manuscript has been judged scientifically suitable for publication and will be formally accepted for publication once it meets all outstanding technical requirements.

Kind regards,

Sonny Shlomo Bleicher, Ph.D.

Academic Editor

PLOS ONE

Additional Editor Comments (optional):

Reviewers' comments:

Reviewer's Responses to Questions

**Comments to the Author**

1. If the authors have adequately addressed your comments raised in a previous round of review and you feel that this manuscript is now acceptable for publication, you may indicate that here to bypass the “Comments to the Author” section, enter your conflict of interest statement in the “Confidential to Editor” section, and submit your "Accept" recommendation.

Reviewer #1: All comments have been addressed

Reviewer #2: All comments have been addressed

2. Is the manuscript technically sound, and do the data support the conclusions?

Reviewer #1: Yes

Reviewer #2: Yes

3. Has the statistical analysis been performed appropriately and rigorously? 

Reviewer #1: Yes

Reviewer #2: Yes

4. Have the authors made all data underlying the findings in their manuscript fully available?

Reviewer #1: Yes

Reviewer #2: Yes

5. Is the manuscript presented in an intelligible fashion and written in standard English?

Reviewer #1: Yes

Reviewer #2: Yes

6. Review Comments to the Author

Reviewer #1: I have repeatedly read the manuscript and cross-referenced the text against the Figures and statistics provided, in addition to selected references, and can find no errors in the data interpretation.

Reviewer #2: (No Response)

7. PLOS authors have the option to publish the peer review history of their article (what does this mean?). If published, this will include your full peer review and any attached files.

Reviewer #1: No

Reviewer #2: No

---

## [Editor Report · Acceptance letter]

10 Jan 2025

PONE-D-24-37502R1 

PLOS ONE

Dear Dr. Major, 

I'm pleased to inform you that your manuscript has been deemed suitable for publication in PLOS ONE. Congratulations! Your manuscript is now being handed over to our production team.

Kind regards, 

on behalf of

Dr. Sonny Shlomo Bleicher 

Academic Editor

PLOS ONE